# Diffusion Feature Field for Text-based 3D Editing with Gaussian Splatting

**Eunseo Koh**      **Sangeek Hyun**      **MinKyu Lee**
**Jiwoo Chung**      **Kangmin Seo**      **Jae-Pil Heo**[†]

Sungkyunkwan University
{colorrain, hsi1032, bluelati98, wldn0202, skmskku, jaepilheo}@skku.edu

## Abstract

Recent advances in text-based image editing have motivated the extension of these techniques into the 3D domain. However, existing methods typically apply 2D diffusion models independently to multiple viewpoints, resulting in significant artifacts, most notably the Janus problem, due to inconsistencies across edited views. To address this, we propose a novel approach termed DFFSplat, which integrates a 3D-consistent diffusion feature field into the editing pipeline. By rendering and injecting these 3D-consistent structural features into intermediate layers of a 2D diffusion model, our method effectively enforces geometric alignment and semantic coherence across views. However, averaging 3D features during the feature field learning process can lead to the loss of fine texture details. To overcome this, we introduce a dual-encoder architecture to disentangle view-independent structural information from view-dependent appearance details. By encoding only the disentangled structure into the 3D field and injecting it during 2D editing phase, our method produces semantically and multi-view coherent edited images while maintaining high quality editing. Additionally, we employ a time-invariance objective to ensure consistency across diffusion timesteps, enhancing the stability of learned representations. Experimental results demonstrate that our method achieves state-of-the-art performance in terms of CLIP similarity, and better preserves structural and semantic consistency compared to existing approaches.

## 1 Introduction

Recent advances in text-based image generation models [28, 29, 27, 22] have enabled their widespread application, particularly in text-based image editing tasks [35, 10, 1, 14]. Driven by increasing demand for intuitive and efficient editing of 3D content, recent efforts have focused on extending these text-based editing techniques from the 2D domain into 3D representations. Consequently, numerous text-based 3D editing [9, 13, 30, 43, 4, 37, 3] approaches have been proposed, significantly broadening their applicability across diverse fields, including film production, game development.

The primary objective of text-based 3D editing is to modify a given source 3D representation according to user-provided textual instructions. To edit a source 3D representation based on the given textual instruction, existing approaches typically follow a two-step process. First, 2D editing methods are applied to multi-view images rendered from the source 3D representation. Subsequently, the 3D representation is optimized to align with these edited 2D reference views.

---

[†]Corresponding author

39th Conference on Neural Information Processing Systems (NeurIPS 2025).

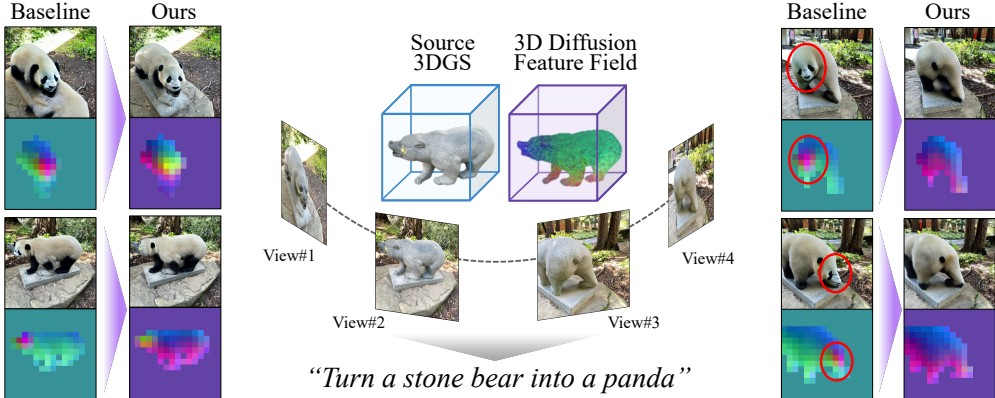

Figure 1: Diffusion-based 2D editing exhibits multi-view artifacts such as Janus problem, visible in the model's intermediate features, that lead to inconsistent edits across viewpoints (referred to as *Baseline*). For example, independent editing of 2D multi-view images causes the panda's face incorrectly appear on the bear's back. By enforcing feature-level multi-view and semantic consistency through our 3D diffusion feature field, these errors are resolved, yielding coherent edits across all views (referred to as *Ours*).

In 2D editing process, achieving *multi-view consistency* across edited images is essential. Multi-view consistency ensures that the edited scenes remain visually coherent across different viewpoints; inconsistent multi-view images inevitably introduce artifacts in the reconstructed 3D scene. However, existing text-based 3D editing methods utilize 2D diffusion models based editing approaches [1] that independently process each view, making it challenging to maintain multi-view consistency.

Here, we focus on the notable artifacts in 3D editing and generation known as the Janus problem [12, 24, 36]. Due to the inherent frontal-view bias in 2D diffusion models, frontal-view semantics attributes are often incorrectly synthesized onto multiple *non*-frontal viewpoints. This issue leads the semantic information to be misaligned in the multi-view images of source 3D during the 2D editing process. Interestingly, we observe that multi-view artifacts, like a panda's face on its back, already emerge in the diffusion model's intermediate features, highlighting that enforcing feature-level consistency is crucial to prevent such errors in the final 3D edits, as shown in Figure 1.

To this end, we propose DFFSplat, which augments the 2D editing phase with a multi-view consistent diffusion feature. Specifically, we aim to build a unified 3D feature field of the source scene that is shared across multiple views, and inject the rendered feature during the 2D diffusion editing [1]. Our key insight is that guiding the diffusion process with the rendered features already coherent in 3D encourages both geometric alignment across views and semantic fidelity to the source content.

However, as 3D features are fused across multiple viewpoints, they converge to an averaged representation and may lose per-view texture details. Accordingly, we observe that directly injecting these averaged features results in blurred edits. From this design, we hypothesize that the information lost due to its view-dependent nature during 3D embedding primarily related to appearance, whereas the preserved view-independent information retained in the feature field relates to the semantic structure.

To address the issue of blurry artifacts, we propose isolating only the view-independent semantic information into the Gaussian Splatting [15] representation, while preserving the remaining view-dependent features directly from the editing process guided by the textual instruction. Given the strong correlation between structural layouts and semantic maps, we anticipate that such an isolation strategy will effectively maintain per-view semantic consistency and simultaneously enhance the visual quality. Finally, we fuse edited appearance features with rendered 3D-consistent semantic features and inject them into selected intermediate layers of the diffusion model.

Experimentally, our method achieves the state-of-the-art CLIP similarity score. More importantly, as shown in Figure 1, it effectively mitigates semantic distortions such as the Janus problem, preserving source identities across views. Additionally, we quantify semantic consistency by measuring the preservation of feature correspondences between image pairs before and after 3D editing, where our method achieves the highest correspondence score among existing approaches.

## 2 Related work

**Diffusion-based text-based editing**    Recently, with the rapid progress of text-based image generation, text-based image editing has also advanced significantly. DiffusionCLIP [17] integrates CLIP-based optimization to refine edits while preserving identity. Prompt-to-Prompt (P2P) [10] introduces a method that leverages source attention maps to guide cross-attention manipulation, enabling precise and structure-preserving modifications through prompt changes. Plug-and-Play (PnP) [35] presents a framework that injects spatial features from a source image to achieve controlled image-to-image translation guided by text prompts while preserving the source layout. Recent approaches like InstructPix2Pix [1] fine-tune models on synthetic paired datasets generated using Prompt-to-Prompt [10], allowing diverse and complex edits through a single forward pass.

Building on text-guided 2D editing techniques, most existing 3D editing approaches operate by applying edits independently to per-view renderings of the source scene, which often leads to multi-view inconsistencies. In contrast, we learn a geometry-aligned, 3D consistent feature field and inject its rendered features during the editing process, thereby enforcing consistency across viewpoints.

**Text-based 3D scene editing**    Many existing text-based 3D scene editing methods build upon text-based 2D image editing techniques. Instruct-NeRF2NeRF [9] applies iterative view-by-view editing using a 2D diffusion model, followed by NeRF [21] optimization. Subsequent studies [32, 5, 40] have proposed various approaches to improve multi-view consistency and enhance the quality of 3D scene editing.

More recently, with the emergence of 3D Gaussian Splatting (3DGS) as a new 3D representation characterized by its remarkably fast rendering speed, several methods have been introduced to enable text-based editing on 3DGS. GaussianEditor [4] proposes a 3D Gaussian splatting editing framework enabling fast and controllable object manipulation through semantic tracing and hierarchical representations. Subsequent works such as Direct Gaussian Editor (DGE) [3] further improved multi-view consistency through geometry-aware edits, however, it is limited to adjacent views. The works of GaussCtrl [38] and VcEdit [37] propose novel methods to enforce multi-view consistency, however not considering the semantic consistency across views. Our DFFSplat framework addresses these limitations by explicitly encoding structural consistency within a learned 3D diffusion feature field, significantly improving semantic and multi-view coherence.

## 3 Preliminary

### 3.1 Representation of feature field in 3DGS

Recent active research on 3D scene understanding [18, 6, 16, 19, 20, 31, 39, 33, 8] aims to embed semantic features into 3DGS [25, 42, 26] by assigning learnable semantic vectors to each Gaussian and training them to match ground-truth semantic maps. The rendering process of semantic features can be formulated analogously to that of RGB images $\mathbf{I} \in \mathbb{R}^{3 \times H \times W}$, where $H, W$ represent the height and width of the image size, as follows:

$$C = \sum_{i \in \mathcal{G}} c_i \alpha_i \mathcal{T}_i, \quad F = \sum_{i \in \mathcal{G}} f_i \alpha_i \mathcal{T}_i, \tag{1}$$

where $\mathcal{G}$ is the ordered set of Gaussians contributing to a given pixel. $C$ and $F$ are the rendered RGB color and semantic feature vector, and $c_i$ and $f_i$ are color and semantic features of the $i$-th Gaussian, respectively. $\alpha_i$ is the product of a learnable opacity parameter and the Gaussian's projected density at the pixel, and $\mathcal{T}_i$ is the accumulated transmittance up to $i$-th Gaussian, defined as $\mathcal{T}_i = \prod_{j=1}^{i-1}(1 - \alpha_j)$. Unlike RGB values $c$, which are view-dependent and thus require spherical harmonic (SH) coefficients conditioned on viewing direction, semantic features $f$ are inherently view-independent. This eliminates the need for additional SH coefficients during the rendering.

In addition, recent practices [25] utilize a low-dimensional semantic latent compressed by an autoencoder as ground-truth semantic features for reducing memory consumption. This process can be formulated as below:

$$Z = E(F), \quad \hat{F} = D(Z), \tag{2}$$

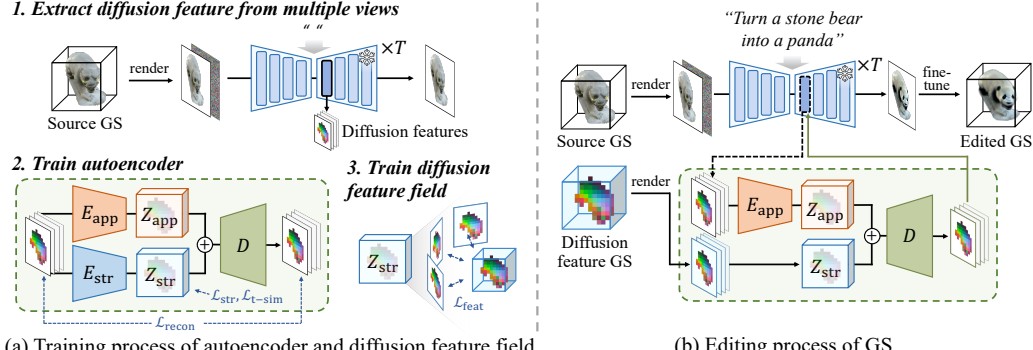

(a) Training process of autoencoder and diffusion feature field      (b) Editing process of GS

Figure 2: Overview of the proposed method. (a) To obtain 3D-consistent diffusion features, we train a 3D diffusion feature field. However, rendering these features for editing averages out per-view details and sacrifices fine textures. To prevent this, we introduce a dual-encoder scheme that decomposes structural and appearance features at the autoencoder level. Then, we train the feature field exclusively on these structural features, which have multi-view consistent geometry. (b) For 3D editing, we edit the multi-view source images by injecting the rendered structural features from our learned feature field. By keeping the appearance feature from images being edited, the results remain 3D consistent while faithfully reflecting the user-given edit text.

where $E(\cdot), D(\cdot)$ are encoder and decoder, and $Z, F, \hat{F}$ are their compressed semantic features, uncompressed and reconstructed counterparts, respectively. The training objective for the autoencoder minimizes the distance between reconstructed and ground-truth features, while the 3DGS objective minimizes the distance between rendered and compressed features.

# 4   3D Gaussian Splatting editing via diffusion feature field

In this paper, we focus on addressing the inconsistencies frequently encountered in existing text-based 3D editing approaches, such as the Janus problem, where independently edited multi-view images fail to maintain consistent geometry and semantics.

To overcome these issues, we propose a novel editing framework that constructs a 3D-consistent diffusion feature field, which is then rendered and injected into the editing process to ensure multi-view and semantic consistency in the edited images. Furthermore, we observe that directly using the learned 3D features in editing causes blurriness in the edited images, as its per-view appearance information are averaged while building 3D feature field. To resolve this, we introduce a dual-encoder architecture that separates structural features from appearance information, and only embeds multi-view consistent structural information within the 3D feature field. Finally, we elaborate how our 3D-consistent structural features are integrated into the 2D diffusion editing pipeline and the subsequent 3D reconstruction process.

## 4.1   3D-consistent diffusion feature field for semantic consistent editing

Diffusion features inherently encode structural semantics within their intermediate representations [35], enabling explicit control over the layout of generated images during editing. Therefore, by constructing diffusion features that are coherent across multiple views (i.e., 3D-consistent), directly injecting these coherent features into a per-view 2D diffusion editing model can effectively produce geometrically view-consistent edited images.

To achieve this, we first construct a diffusion feature field by aligning and fusing per-view diffusion features extracted from multi-view images of the source 3D scene. Once trained, this feature field provides 3D-consistent diffusion features, which we render and inject at key layers within the diffusion model. This explicitly encourages geometric consistency across multi-view images and semantic fidelity to the original content from source throughout the editing process. This strategy proves effective because diffusion features extracted from multi-view images inherently possess a degree of semantic coherence. Moreover, this 3D diffusion feature field also facilitates the correction of inconsistent or semantically biased features that may arise in individual views. Such inconsistencies

are typically (1) view-dependent and (2) relatively sparse across the entire set of viewpoints. Therefore, by explicitly averaging structural information across multiple views, the 3D feature field can correct biased views by utilizing structural information from unbiased views. This capability of cross-view averaging offered by the 3D feature field effectively resolves these biases.

Following prior work [25, 42, 26], we first train the 3D diffusion feature field on the source 3DGS, as detailed in Sec. 3.1. However, diffusion features differ notably from typical vision model features due to the presence of a timestep $t$. Thus, we construct a timestep-conditioned encoder $E_{\text{base}}$ which has inputs as a diffusion feature of specific timestep $F^t$ and its corresponding timestep $t$. This process can be formulated as below:

$$Z_{\text{base}}^t = E_{\text{base}}(F^t, t) \tag{3}$$

where $Z_{\text{base}}^t$ is compressed latent of $F^t$ by the encoder $E_{\text{base}}$. However, training 3D feature field based on this naïve encoder has several issues as noted in below section, so we introduce a dual-encoder scheme.

## 4.2 Dual-encoder scheme for decomposing structure-appearance features

By rendering features from our 3D diffusion feature field, we inherently obtain 3D-consistent features. However, aggregating information across multiple viewpoints causes per-view appearance details to become averaged out. Consequently, we observe that directly injecting these averaged features during editing results in blurred outputs with insufficient fine textures. This averaging issue is further exacerbated by the inherently time-variant nature of diffusion features during the denoising process, making it challenging to accurately represent these features within conventional 3D feature field. Consequently, directly injecting these averaged features during editing results in blurred outputs lacking sufficient fine textures. Although such representations might be acceptable in 3D recognition tasks [25, 42], which only need view-independent semantic information, they significantly limit editing applications, where preserving visual detail and fidelity to user-provided instruction is crucial.

To address this limitation, we explicitly separate appearance information (fine, per-view textures, and time-variant) from structural information (geometric, viewpoint-consistent, and time-invariant). We first train the diffusion feature field of 3DGS exclusively on structural information to ensure robust geometric consistency across views. During editing, we fuse the rendered features from this field with appearance information specifically generated to align with user-provided prompts. This combined representation enables our method to produce semantic-consistent edits, accurately reflecting user inputs while effectively preserving viewpoint-specific appearance details.

To this end, we propose a dual-encoder scheme that separates appearance and structural features at the encoder level. Specifically, the encoder is divided into two distinct branches: a structure encoder $E_{\text{str}}$ and an appearance encoder $E_{\text{app}}$. Following conventional autoencoder setups, the outputs from these two encoders are combined and subsequently processed through a shared decoder for reconstruction. This process can be formally described as follows:

$$Z_{\text{app}}^t = E_{\text{app}}(F^t, t), \quad Z_{\text{str}}^t = E_{\text{str}}(F^t, t), \quad \hat{F}^t = D(Z_{\text{app}}^t, Z_{\text{str}}^t), \tag{4}$$

where $Z_{\text{str}}^t$ and $Z_{\text{app}}^t$ are structural and appearance features encoded by corresponding encoders, respectively.

Then, in training phase, we explicitly differentiate the roles of the two encoders by imposing an additional constraint features to learn on the latent representation of the structure encoder. Specifically, we leverage the self-similarity maps derived from DINOv2 [2, 23] features, which have demonstrated effectiveness in capturing structural information [34]. These self-similarity map serve as an auxiliary objective by minimizing their difference from the self-similarity maps generated from the structure encoder's latent output. Due to the inherent information-bottleneck characteristic of autoencoders, enforcing structural constraints within the structure encoder naturally encourages the complementary appearance encoder to capture per-view, appearance-specific details. This structure similarity loss is formulated as follows:

$$\mathcal{L}_{\text{str}} = \left\| S(F_{\text{dino}}(\mathbf{I})) - S(Z_{\text{str}}^t) \right\|_2, \text{where } S(K)_{ij} = \text{cos-sim}(K_i, K_j), \tag{5}$$

where $t$ is sampling from $\mathcal{I} \sim (T/n, 2T/n, \ldots, T)$ where $T$ is the number of diffusion timesteps and $n$ is the number of timestep interval, cos-sim$(\cdot, \cdot)$ is the cosine similarity between two feature maps and $S(K) \in \mathbb{R}^{hw \times hw}$ is a self-similarity map of a given feature map $K \in \mathbb{R}^{c \times hw}$, where

$i, j \in \{1, ..., hw\}$. By leveraging structure information extracted from the rendered image $\mathbf{I}$ of the source 3D scene using a DINOv2 extractor $F_{\text{dino}}$, $E_{\text{str}}$ effectively captures semantic information. To match the spatial resolution with DINOv2 features, we interpolate the DINOv2 features to the same resolution as the structure encoder's latent output.

We further enforce that the structure encoder's latent outputs at different timesteps remain consistent, ensuring they can be represented by a single, static 3D scene. These constraints can be formulated as below:

$$\mathcal{L}_{\text{t-sim}} = \frac{1}{n(n-1)} \sum_{t,t' \in \mathcal{I}} \mathbf{1}_{[t' \neq t]} \left\| Z_{\text{str}}^t - Z_{\text{str}}^{t'} \right\|_2. \tag{6}$$

$\mathcal{L}_{\text{str}}$ is a structure similarity loss that imposes the latent of structure encoder $Z_{\text{str}}$ to have structural information mostly, and $\mathcal{L}_{\text{t-sim}}$ is a time-similarity loss that encourages the structure encoder's latent representations to be invariant across diffusion timesteps.

**Training 3D diffusion feature field**    As shown in Figure 2-(a), we train our 3D diffusion feature field using the dual-encoder scheme described in Sec. 3.1. Concretely, we optimize an autoencoder with three loss terms: (1) a standard reconstruction loss to recover per-view diffusion features, (2) a structural similarity loss $\mathcal{L}_{\text{str}}$ based on DINOv2 self-similarity maps, and (3) a time-similarity loss $\mathcal{L}_{\text{t-sim}}$ to enforce invariance across diffusion timesteps. In addition, we assemble the structure encoder's latent outputs $Z_{\text{str}}^t$ over all timesteps $t \in [0, T]$ for each source view, average them to form a static 3D target feature, and render this target via our 3DGS parameterization. We then train the 3DGS parameters $f$ by minimizing the feature-map discrepancy between the rendered feature map $\tilde{F}$ and the time-averaged target $\bar{Z}_{\text{str}}$, thereby grounding our 3D field in multi-view, time-invariant structural information. This process can be formulated as below:

$$\mathcal{L}_{\text{ae}} = \mathcal{L}_{\text{recon}} + \lambda_1 \mathcal{L}_{\text{str}} + \lambda_2 \mathcal{L}_{\text{t-sim}}, \quad \mathcal{L}_{\text{recon}} = \frac{1}{T} \sum_{t \in [0,T]} d(\hat{F}^t, F^t), \tag{7}$$

$$\mathcal{L}_{\text{feat}} = d(\tilde{F}, \bar{Z}_{\text{str}}), \quad \text{where} \quad \bar{Z}_{\text{str}} = \frac{1}{T} \sum_{t \in [0,T]} Z_{\text{str}}^t, \tag{8}$$

where $d(\cdot, \cdot)$ is a distance function (e.g., L2 distance), and $\mathcal{L}_{\text{ae}}$ and $\mathcal{L}_{\text{feat}}$ are objectives of autoencoder and feature of 3DGS, respectively, and $\lambda_1, \lambda_2$ are the strengths of regularization for structure and time-invariance.

### 4.3    3D Gaussian Splatting editing via view-consistency guidance

In this section, we describe how the trained diffusion feature field of 3DGS and the dual-encoder mechanism are leveraged to effectively obtain multi-view edited images. In particular, given an instruction prompt from the user, our primary goal during editing is to preserve the appearance information corresponding to the edit, while ensuring semantic consistency with the source scene.

To achieve this, we first encode the features obtained during the editing process using the appearance encoder $E_{\text{app}}$. Then, to enforce semantic consistency with the source, we simultaneously render view-consistent semantic features from the 3DGS semantic feature field, thereby providing essential source semantic information. These rendered 3D-consistent semantic features are subsequently fused with the encoded appearance features and then passed through the decoder to obtain reconstructed diffusion features. Finally, by injecting these diffusion features into the original editing pipeline (i.e., by replacing the model's native features with our diffusion features at selected intermediate layers), we produce multi-view edited images that faithfully reflect the user's prompt while accurately preserving the source semantics (Figure 2-(b)). This process is formulated as below:

$$\hat{F}_{\text{injected}}^t = D(E_{\text{app}}(F^t, t), \tilde{F}), \tag{9}$$

where $\hat{F}_{\text{injected}}^t$ is modified diffusion feature during editing, which contains 3D-consistent semantic information and appearance of editing condition.

Because this injection effectively supplies a structure-guided conditioning, we augment the original guidance [11] in InstructPi2Pix [1] with an additional "view-consistency guidance" term. This combined guidance ensures that edits not only follow the user's text instructions but also remain geometrically coherent across all views. This process is defined as below:

$$\begin{aligned}
\hat{e}_\theta(z_t, c_I, c_S, c_T) &= e_\theta(z_t, \phi, \phi, \phi) + s_I \cdot (e_\theta(z_t, c_I, \phi, \phi) - e_\theta(z_t, \phi, \phi, \phi)) \\
&+ \underline{s_S \cdot (e_\theta(z_t, c_I, c_S, \phi) - e_\theta(z_t, c_I, \phi, \phi))} + s_T \cdot (e_\theta(z_t, c_I, \underline{c_S}, c_T) - e_\theta(z_t, c_I, \underline{c_S}, \phi))
\end{aligned} \tag{10}$$

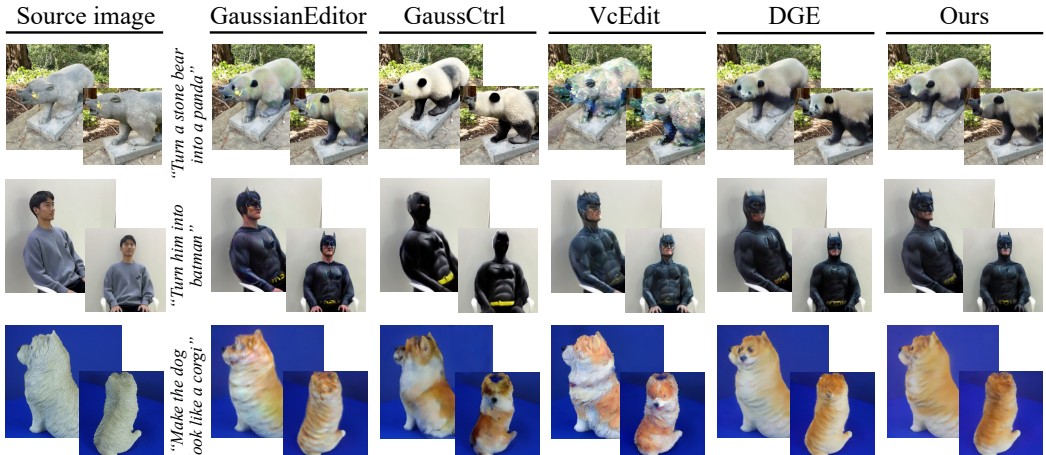

Figure 3: Qualitative comparison of edited 3D scene. While existing methods often suffer from multi-view inconsistencies due to the inherent frontal-view bias of diffusion models, causing severe artifacts in the edited 3DGS outputs, our approach effectively preserves the original source structure and achieves faithful edits aligned with the input text prompt.

where $e_\theta$ and $\hat{e}_\theta$ is a diffusion model and its guided prediction, and $z_t$ is an input noised latent, and $c_I$ and $c_S$ and $c_T$ are the conditions about image and structure and text, and $s_{(\cdot)}$s are a strength of guidance. Underlined notations denote our modification, the "view-consistency guidance" we additionally use.

## 5 Experiments

### 5.1 Implementation details

We conduct the experiments on 5 scenes from IN2N [9] and DreamEditor [43], including complex 360° scenes. To validate the effectiveness of the proposed methods, we compare ours with previous 3DGS editing methods [4, 38, 37, 3]. For a fair comparison, we use InstructPix2Pix as an image editor, which is the most popular editor used by prior methods [4, 3]. During editing, we utilize classifier-free guidance with scales of 7.5 for textual conditions and 1.5 for image conditions, unless specified otherwise. For each scene, we edit 20 views of the source scene and use the edited images as the reconstruction target to update 3DGS. For training details of 3DGS, we iteratively update the parameters of 3DGS for a total of 1000 iterations, while image editing is performed every 500 iterations. We use Adam optimizer to update the 3DGS parameters and LPIPS [41] and L1 loss as reconstruction objectives, following IN2N [9]. For training our dual-encoder, we set $\lambda_1 = 1.0$ and $\lambda_2 = 0.05$. We further elaborate on the additional implementation and details in supplementary.

**Evaluation metric**  Following the previous baselines [9, 37, 3], we employ CLIP similarity and CLIP directional similarity [17, 7] as evaluation metrics for quantitative comparison. CLIP similarity is computed as the cosine similarity between text and image embeddings, and clip directional similarity measures the difference in embedding directions from the target embedding to the source embedding between the image and text modalities. Both metrics evaluate how well the edited 3DGS reflects the target prompt by comparing image and text embeddings, but clip directional similarity is known for its robustness to mode collapse. We use every image available in the training dataset for evaluation, although only use 20 views of images for editing.

### 5.2 Comparisons with previous works

**Quantitative comparison with previous works**  We first compare CLIP and its directional similarity to measure the text fidelity of the edited scene. As reported in Table 1, the proposed method achieves significantly higher scores in both CLIP and clip directional similarity. Furthermore, ours reports the lowest editing time, similar to prior arts [3].

Table 1: Quantitative comparison. CLIP and clip dir. denotes CLIP and clip directional similarities, respectively.

| Method | CLIP | clip Dir. | Avg. Time |
|---|---|---|---|
| GaussianEditor [4] | 24.40 | 16.67 | ∼7min |
| GaussCtrl [38] | 22.87 | 13.17 | ∼8min |
| VcEdit [37] | 23.18 | 14.03 | ∼20min |
| DGE [3] | 24.66 | 17.21 | ∼4min |
| Ours | **25.11** | **18.15** | ∼4min |

Table 2: Ablation study of proposed components and 3D feature field, showing complementary gains in text–image alignment.

| Method | CLIP | CLIP Dir. |
|---|---|---|
| Ours (Full method) | **25.11** | **18.15** |
| - $\mathcal{L}_{\text{t-sim}}$ | 25.09 | 18.09 |
| - $E_{\text{app}}$ | 24.80 | 17.21 |
| Ours (Full method) | **25.11** | **18.15** |
| - 3D feature field | 25.00 | 17.82 |

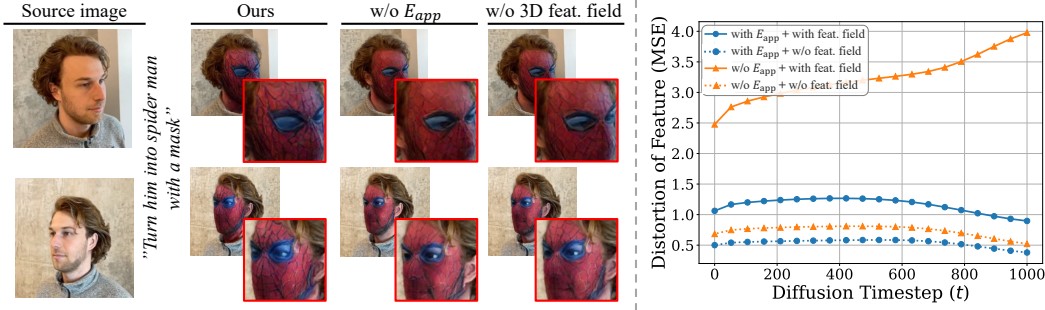

Figure 4: Ablation study. (Left) Qualitative comparison of ablated models. (Right) Measuring distortion between the reconstructed diffusion feature ($\hat{F}^t$) and the modified diffusion feature ($\hat{F}^t_{\text{inject}}$) across ablated models. Lower distortion indicates better preservation of source representation. The presence of an appearance encoder ($E_{\text{app}}$) reduces the distortion introduced by the 3D diffusion feature field ("feat. field" in the legend), demonstrating our dual-encoder effectively preserves information.

**Qualitative comparison with previous works**    Qualitative results are shown in Figure 3. The proposed method produces much fewer artifacts in the rendered images from edited scene. In particular, we observe the edited results from ours achieve a prominent semantic consistency, preserving the composition and semantics of the source object. It proves the effectiveness of learning the underlying structure of the source 3D scene via our 3D diffusion feature field.

For example, in the "Turn a stone bear into a panda" instruction (1[st] row), competing methods (e.g., VcEdit, DGE) fail to correct the diffusion model's frontal-view bias, resulting in multi-view inconsistencies that synthesize visible artifacts in the edited 3DGS. In contrast, ours successfully edit the 3D scene without such artifacts, demonstrating the coherence of edited images across all views. Moreover, in the "Turn him into Batman" prompt (2[nd] row), our approach better preserves the source geometry than DGE, demonstrating stronger semantic fidelity to the original 3D scene. Finally, as shown in the last row, unlike other methods that exhibit strong frontal bias, our approach does not display such bias even on the backside of the object.

### 5.3    Analysis on the proposed components

**Ablation study**    We perform an ablation study on our key proposed components, specifically the dual-encoder scheme with structural loss ($E_{\text{app}}$ and $\mathcal{L}_{\text{str}}$) and the time-invariance objective ($\mathcal{L}_{\text{t-sim}}$). Note that a naïve decomposition of encoders without the structural loss does not effectively separate learned representations; thus, we ablate these components together.

As reported in Table 2, the dual-encoder scheme substantially impacts performance, improving the clip directional similarity. This emphasizes the importance of explicitly decomposing structural and appearance information to achieve detailed, text-consistent 3D edits. Visual examples comparing our final model to its ablations are provided on the left side of Figure 4. Without the structure decomposition, the edited results lose per-view details, becoming overly blurred due to averaged features during editing.

We also ablate the time-invariance objective $\mathcal{L}_{\text{t-sim}}$ and quantitatively confirm that it boosts both the CLIP similarity and directional scores. Additionally, we evaluate the significance of learning the 3D

Table 3: Difference in correspondence matching map between multi-view source and edited images from complex 360° scenes. Our method demonstrates superior editing performance under sparse-view conditions compared to others.

| Method | Bear scene | Dog scene |
|---|---|---|
| GaussianEditor [4] | 5.08 | 8.01 |
| GaussCtrl [38] | 10.91 | 8.15 |
| VcEdit [37] | 6.02 | 6.59 |
| DGE [3] | 4.14 | 7.36 |
| Ours | **3.93** | **5.53** |

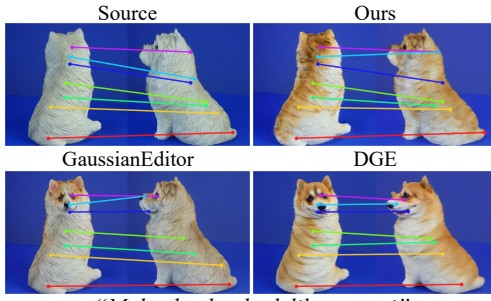

*"Make the dog look like a corgi"*

Figure 5: Correspondence matching comparison between two edited views of the dog scene.

diffusion feature field itself. That is, we directly inject the original 2D feature into the editing process, instead of rendering the 3D feature field. As shown in Table 2, omitting this field ("-3D feature field" row) significantly decreases text fidelity, as the edited images lose multi-view consistency and appear more blurred during 3D reconstruction process. Qualitative comparisons in Figure 4 further illustrate the necessity of the learned 3D field for achieving detailed textures.

Finally, we measure the distortion caused by feature injection in the reconstructed diffusion features ($\hat{F}^t_{\text{inject}}$) across our ablated models. Note that, we give null text as an editing condition for only validating the distortion caused by feature injection, not the editing capability. The lower distortion indicates the injected feature $\tilde{F}$ does not lose its fine-details, as it can recover these details at the reconstructed feature $\hat{F}^t_{\text{inject}}$. As illustrated in Figure 4, we conducted an ablation study on the presence or absence of the dual-encoder scheme (specifically, the appearance encoder, $E_{\text{app}}$) and the 3D diffusion feature field. The presence of $E_{\text{app}}$ reduces the distortion caused by the 3D diffusion feature field, demonstrating that our dual-encoder effectively preserves information. Additionally, in the case without $E_{\text{app}}$ but with the 3D diffusion feature field, the distortion increases as the timestep (noise-level) increases, indicating that the 3D diffusion feature field struggles to accurately retain time-invariant noise information.

**Evaluating multi-view and semantic consistency via correspondence matching**  Previous experiments mainly evaluated fidelity to the input text prompts. Here, we further assess how well the edited results preserve the original structure and resolve known editing artifacts, such as the Janus problem. To achieve this, we compute pixel-level correspondence matching maps among multi-view images rendered from both the original and edited 3D scenes, and then measure the difference between these maps. A smaller difference indicates better preservation of the source scene's geometry and improved multi-view consistency due to accurate pixel-level matching across views.

Quantitative results for correspondence matching differences are presented in Table 3. As shown, our proposed method achieves significantly lower distortion in correspondence matching compared to baselines. Qualitative visualizations of the correspondence maps are provided in Figure 5, where baseline methods show noticeable artifacts (e.g., generating faces on the back of an animal), indicating failures in preserving source structure. In contrast, our approach consistently preserves the original structure, resulting in highly accurate correspondence maps.

**Analyze the impacts of 3D feature field via correspondence matching**  To further analyze the individual impacts of DINO regularization ($\mathcal{L}_{\text{str}}$) and the 3D feature field, we conducted the correspondence matching experiment by selectively removing each component. Specifically, we compared scenarios without DINO regularization and without the 3D feature field, respectively. Since the appearance encoder cannot be trained effectively without DINO regularization, we performed the comparative experiments excluding the appearance encoder. In Table 4, we observe that excluding the 3D feature field results in a significantly larger difference in the correspondence matching map compared to excluding DINO regularization. This clearly indicates that the 3D feature field substantially contributes to improving multi-view consistency and semantic consistency, rather than DINO regularization.

Table 4: Correspondence matching score without DINO regularization or 3D feature field.

| Method | Bear scene | Dog scene |
|---|---|---|
| w/o $E_{app}$ | 2.96 | 5.28 |
| w/o $E_{app}$ + w/o DINO regularization | 3.02 | 5.21 |
| w/o $E_{app}$ + w/o 3D feature field | 3.15 | 6.10 |

*"Make the bear look like a grizzly bear"*

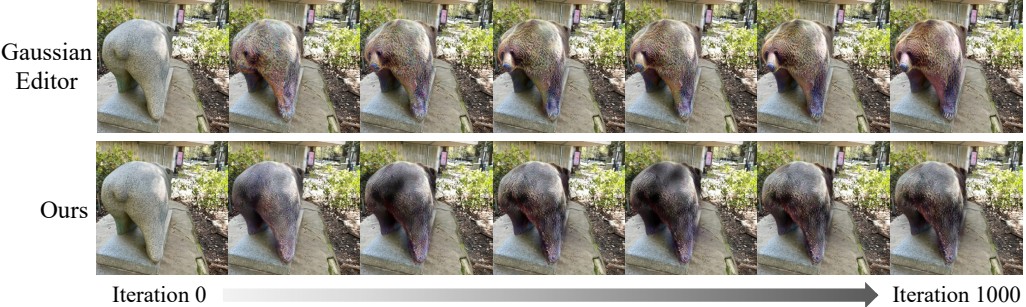

Figure 6: Edited 3DGS results across iterations. While our baseline, GaussianEditor, suffers from the Janus problem, our method effectively mitigates it.

**Visualization of the edited 3DGS over optimization iterations**   We additionally visualize the evolution of the edited 3DGS over the course of optimization iterations. As shown in Figure 6, our baseline method (GaussianEditor [4]) suffers from the Janus problem, where undesired facial attributes emerge on the backside of the bear at very early iterations and gradually intensify as optimization progresses. In contrast, our proposed method effectively suppresses such artifacts throughout the optimization process, maintaining consistent and faithful editing results across all viewpoints.

## 6   Limitation

Our proposed editing framework is inherently constrained by the editing capability of Instruct-Pix2Pix; therefore, it cannot handle 3DGS editing cases that go beyond the editing scope defined by InstructPix2Pix. This dependency fundamentally limits the diversity and complexity of the edits that can be achieved within our framework. Furthermore, since our method places a strong emphasis on preserving the structural integrity of the source 3DGS, it is less suitable for tasks that require substantial geometric transformations. As a result, editing scenarios involving large pose changes, object repositioning, or major structural alterations remain challenging. Addressing these limitations, either by incorporating more flexible editing modules or by enabling geometry-aware transformations, will be an important direction for our future work.

## 7   Conclusion

We introduced DFFSplat, a novel framework for text-based 3D editing that significantly improves multi-view and semantic consistency by incorporating a 3D-consistent diffusion feature field. By leveraging a dual-encoder architecture to disentangle structural and appearance information, along with enforcing time-invariance across diffusion steps, our method effectively addresses prevalent artifacts such as the Janus problem. Experimental results confirm that our approach outperforms prior methods in terms of CLIP similarity and semantic coherence, highlighting its potential for high-quality 3D content editing.

## Acknowledgements

This work was supported in part by MSIT/IITP (No. RS-2022-II220680, RS-2020-II201821, RS-2019-II190421, RS-2024-00459618, RS-2024-00360227, RS-2024-00437633, RS-2024-00437102, RS-2025-25442569), MSIT/NRF (No. RS-2024-00357729), and KNPA/KIPoT (No. RS-2025-25393280).

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
