# OpenReview forum: "Diffusion Feature Field for Text-based 3D Editing with Gaussian Splatting"
_NeurIPS.cc/2025/Conference — NeurIPS 2025 poster_

### Official Review · Reviewer_WiaQ · 2025-06-30

**Clarity:** 3
**Significance:** 3
**Originality:** 3
**Rating:** 3
**Confidence:** 4

**Summary:**

This paper presents a method for 3D editing using 3D diffusion features. The authors mainly want to deal with the view consistent problem during editing, i.e., current methods use single-view diffusion model for editing and lack 3D constraint and may lead to Janus problem. The authors deal with this issue by training a 3D consistent feature field that disentangles the structure information (view independent) and appearance information (view dependent). During editing, the diffusion process is conditioned on the rendered structure features to ensure 3D consistency. The authors perform comparison and proved that their methods can outperform baselines.

**Questions:**

I think the paper is limited on structural editing, which is hard to address in the current framework. I would like to see authors how to address this during rebuttal and adjust my score accordingly.

**Ethical Concerns:**

["NO or VERY MINOR ethics concerns only"]

**Final Justification:**

I am still negative towards this paper due to lack of video results (which is crucial for 3D and video tasks) and missing experiments of structural editing. The overall idea is great, but lack of these prevents me from accepting.

**Limitations:**

Yes

**Paper Formatting Concerns:**

No.

**Quality:**

3

**Strengths And Weaknesses:**

Strengths:
- This paper is well motivated and technically sound. The authors started by analyzing the Janus issue of 3D editing methods, then proposed use 3D consistent features to deal with the issue, following a feature disentanglement training strategy.
- The proposed method is easy to understand and follow. The authors provided relevant equations, figures and details for readers. With the clear motivation, the reviewers are able to understand the whole process of the proposed method.
- The ablation study and detailed analysis of the proposed method is valuable and improves the understanding, including Fig. 4 the diffusion timesteps part, the correspondence matching metrics. This helps the readers understand the effectiveness of this method and its underlying reasons.
- The proposed method, although is optimization based, is quite efficient and only takes 4 minutes and one GPU.

Weaknesses:
- One major and very important limitation of this method is that it cannot perform editing with structure changes, such as deletion, adding etc, since it relies on the 3D structure features during editing that are trained from input multi-view images. The authors also haven't show any examples on this. Therefore, this limits the functionality of 3D editing.
- I would also like to see more results of video comparisons, which can help readers better spot the differences between different methods.

---

> ### Author Rebuttal · Authors · 2025-07-31
>
> ## [@WiaQ] - Is 3D editing with structure changes possible?
>
> We appreciate your valuable feedback regarding the limitations related to structural editing, such as object removal and incorporation. Our method builds upon the GaussianEditor [1] framework, which inherently supports these types of structural changes.
>
> For object removal, GaussianEditor identifies the Gaussian representations for removal using masks and subsequently removes these Gaussians. However, simply removing these Gaussians creates empty regions, necessitating additional steps to fill these areas appropriately. GaussianEditor addresses this by employing a 2D inpainting technique guided by precise masks. To accurately determine these masks, GaussianEditor leverages a K-Nearest Neighbor (KNN) algorithm to identify Gaussians at the boundaries near the removed objects, thereby achieving precise 3D inpainting regions. These boundary Gaussians are projected across multiple views to obtain accurate multi-view 2D masks. Consequently, the method can effectively fill the removed regions with appropriate content guided by these masks.
>
> However, relying solely on single-view masking can result in inaccurate masks, especially in regions affected by occlusion from that particular viewpoint. To address this issue, we leverage our 3D feature field, which effectively captures rich semantic information, enabling clearer distinctions between foreground objects and background regions. PCA visualizations of these semantic-rich features demonstrate their effectiveness in accurately separating foreground and background areas. Utilizing these features would enhance the precision of identifying foreground Gaussians and accurately determining regions requiring inpainting across multiple views. We will include detailed evidence supporting this improvement in the supplementary materials. Nevertheless, we agree with your insight that directly employing structural features during editing may inherently constrain scene structure preservation and complicate object removal; thus, we avoid their direct use for this specific task.
>
> Regarding object adding, GaussianEditor inputs a prompt (P) and a mask (M) corresponding to the target region in an image (I) rendered from a specific viewpoint. Using a 2D inpainting diffusion model, it generates the desired object within the masked region according to the provided prompt. Then, the generated object's foreground is then extracted and transformed into a coarse 3D mesh through an Image-to-3D method, which is subsequently converted into a Gaussian representation and integrated into the scene. However, this direct Gaussian integration often results in unnatural boundaries between the original scene and the added object.
>
> We can address this challenge by utilizing our 3D feature field. Specifically, after generating a scene incorporating the new object, we obtain a diffusion-based feature field for the scene. Since this field encodes semantic information about the added object, the derived structural features effectively guide the editing process, resulting in the addition of the new object without boundaries.
>
> In summary, our method can indeed extend effectively to object removal and incorporation tasks, leveraging our 3D feature field to significantly enhance performance compared to existing methods.
>
> [1] GaussianEditor: Swift and Controllable 3D Editing with Gaussian Splatting
>
> ## [@WiaQ] - More Results of Video Comparisons
>
> Regarding your request for additional video comparison results, we agree that further video examples would greatly aid readers in visually distinguishing differences between methods. We will include more comprehensive video comparisons in the supplementary materials of our final manuscript to clearly illustrate the advantages and limitations of our approach relative to existing methods.

---

> > ### Comment · Reviewer_WiaQ · 2025-08-02
> >
> > I would like to thank the author for demonstrating the potential of structural editing. Personally, I think this is an important feature, so missing that in the original paper would make this paper not complete. I am inclined to remain my score considering video results also missing. I am happy to join the discussion if other reviewers found this paper is acceptable even with these limitations.

---

> > > ### Author Response · Authors · 2025-08-04
> > >
> > > We sincerely thank the reviewer for recognizing the importance and potential of structural editing. Unfortunately, due to rebuttal limitations, we were unable to include video results demonstrating our approach, which we deeply regret. We will thoroughly address your feedback by enhancing and expanding the structural editing results and incorporating comprehensive video evaluations in the revised manuscript.

---

### Official Review · Reviewer_LEXJ · 2025-07-02

**Clarity:** 3
**Significance:** 3
**Originality:** 2
**Rating:** 4
**Confidence:** 4

**Summary:**

Diffusion Feature Field for Text-based 3D Editing with Gaussian Splatting presents a novel approach for multi-view consistent editing of 3DGS. It particularly focuses on the multi-head Janus problem when using image based editing and proposes to disentangle structure and appearance for more faithful editing. It takes a common approach for rendering 3DGS with diffusion features to make a diffusion feature field of the scene. Its primary contribution is a dual encoder scheme where structure and appearcne of the features is encoded separately. It applies self similarity maps of DINO derived from [32] to force the encoder to learn structural information which in turn forces the other encoder to encode apperance as the complete feature set is expected during the fused decoding process. These encodings are carefully injected at specific layers of the editing pipeline (InstructPix2Pix) to ensure structural consistency. The authors further propose to ensure consistency of the diffusion features at different time steps. The method is compared quantitaively against 3 baselines using CLIP similarity and direction metrics and an ablation is conducted.

**Questions:**

* Can the authors clarify when forcing DINO structural features to be similar to diffusion features, how these 2 features relate to each other and if the authors handle varying shapes of the feature maps from the 2 models?
* How do the authors select the specific layers for injecting diffusion features into IP2P?
* Please particularly address points 2 and 3 of the weaknesses

**Ethical Concerns:**

["NO or VERY MINOR ethics concerns only"]

**Final Justification:**

My concerns have been largely addressed and I think this paper has many merits that would be useful to the community but unofortunately I cannot wholeheartedly recommend acceptance (only weak accept) due to the omission of videos in the supplementary to clearly judge the quality of the edited 3DGS

**Limitations:**

While limitations have been discussed from an instructpix2pix perspective, can the authors also discuss limitations of the dual encoding stratergy?

**Quality:**

2

**Strengths And Weaknesses:**

## Strengths
* The paper is quite well-written and well-explained.
* The paper proposes a novel and simple solution for disentangling geometry and appearance in 3DGS using the dual encoding stratergy employing self similarity maps.
* The resuls show the method addresses the problem of multi-head Janus well. The quantitaive results show reasonable gains compared to baselines.

## Weaknesses
* The related work is missing several papers-- Decomposing nerf for editing via feature field distillation, ProteusNeRF, GSEditPro, etc.
* The need for separate DINO structural features. The method forces the diffusion structural features to be similar to DINO structural features (self similarity maps). Why can't the diffusion features be forced to learn structural information directly by utilizing self similarity maps? Moreover, this forcing of diffusion features to be more like DINO can lose out on the geometrical improvements which diffusion features bring over DINO and act as complementary [1]. And if DINO is needed, why not render DINO features directly rather than diffusion features?
* Diffuion features can be extracted from images using a single timestep but this paper utilizes multiple time steps. While this can have some improvements, the paper also forces the diffusion features to be coherent across time steps which seems a bit orthogonal. Can an ablation be conducted to show the need for multi time step diffusion features which is computationally more expensive?
* The variety of scenes shown is quite limited and I find the omission of videos in the supplementary a big limitation to clearly judge the quality of the edited 3DGS (such as floaters).
* Adding GSEDitPro to the baseline would make the evauation stronger


## Suggested improvements:
* The figure captions do not properly explain the figures. For example, in Figure 1, it's not understandable at first glance what the feature maps are trying to convey.
* A few sentences are not grammatically correct such as L125-127.

[1] https://arxiv.org/pdf/2305.15347

---

> ### Author Rebuttal · Authors · 2025-07-31
>
> ## [@LEXJ] - Incomplete Coverage of Related Work
>
> Thank you for pointing out the important omissions in our related work section. We acknowledge that including references such as "Decomposing NeRF for Editing via Feature Field Distillation", "ProteusNeRF", and "GSEditPro" will enhance the comprehensiveness of our discussion. We will incorporate and carefully discuss these additional works, clearly highlighting how our methodology relates to and differs from them, in the revised paper.
>
> ## [@LEXJ] - Redundancy and Drawbacks of Aligning Diffusion Features with DINO
>
> Thank you for raising insightful questions about the necessity of separate DINO structural features. As noted in the [1], "DINO features can capture high-level semantic information and excel at obtaining sparse but accurate matches. In contrast, Stable Diffusion features focus on low-level spatial information and ensure the spatial coherence of the correspondence." Consequently, Stable Diffusion features predominantly capture low-level spatial layout details, whereas DINO features more effectively encode higher-level semantic content.
>
> Given our primary goal of addressing the Janus problem in 3D editing, maintaining consistent semantic information from the source scene is critical. While Diffusion features indeed contain sufficient spatial structure information, their heavy reliance on spatial positioning within images makes them less effective for mitigating semantic inconsistencies such as the Janus issue. In practical tests, PCA visualizations of Diffusion features from different views of an object (e.g., the left and right sides of a bear) revealed that spatial positions within the image significantly influenced feature similarity. This spatial dependency makes Diffusion features inherently less suited for addressing the Janus problem.
>
> To empirically verify the benefit of using DINO features for regularization, we conducted experiments comparing the training of structure features to match self-similarity maps derived from both Diffusion and DINO structural features. The outcomes demonstrated notably superior performance when regularizing with the self-similarity maps of DINO structural features.
>
> We agree your point that using DINO features for regularization might reduce the benefits of the spatial information inherently present in Diffusion features. Additionally, while rendering DINO features directly could be a potential alternative, integrating them directly into the Diffusion generation process poses practical challenges. Exploring methods to effectively combine the complementary strengths of both Diffusion and DINO features represents an intriguing and valuable avenue for future research.
>
> [1] Zhang, Junyi, et al. "A tale of two features: Stable diffusion complements dino for zero-shot semantic correspondence." *Advances in Neural Information Processing Systems* 36 (2023): 45533-45547.
>
> ## [@LEXJ] - Justification for Multi-Timestep Usage
>
> Thank you for your valuable observation regarding our use of multi-timestep diffusion features. We agree that diffusion features could indeed be extracted from images using a single timestep. However, directly substituting features from one timestep to another introduces a problem arising from differing noise levels at each timestep. Injecting features extracted at different noise levels can significantly impact the generative capability of the diffusion model. Therefore, instead of relying solely on single-timestep features, we propose an approach that injects features specifically matched to the appropriate noise level at each timestep.
>
> Recognizing the computational cost of generating separate 3D feature fields for multiple timesteps, we introduced the dual-encoder strategy to address this challenge. Our structure encoder is explicitly designed to extract noise-invariant structural information from the features, while our appearance encoder captures information that includes noise. Thus, the appearance encoder effectively maintains noise-level information across multiple denoising steps during editing. The structure encoder’s noise-invariant features are then rendered through our 3D feature field, allowing consistent structural information to be provided across various timesteps irrespective of their noise levels.
>
> We provide an experimental comparison between training our dual encoder and 3D feature field using single-timestep features (extracted at the last denoising step) versus multi-timestep features, employing the same experimental setup and scenes as described in our main paper. Our results clearly indicate lower CLIP scores and higher MSE values when trained with single-timestep features. The higher MSE specifically suggests increased feature distortion, which we attribute to the appearance encoder’s inability to generalize across different noise levels due to being trained exclusively at a single timestep.
>
> | Method | CLIP | CLIP Dir. | MSE |
> | --- | --- | --- | --- |
> | single timestep | 25.08 | 17.98 | 1.7144 |
> | multi timesteps | 25.29  | 17.69  | 1.0931 |
>
> ## [@LEXJ] - Evaluation Limitations
>
> We agree that incorporating a broader range of scenes and providing video examples will greatly enhance the clarity and assessment of our results. We will address these points by adding more diverse scenes and supplementary videos in future. Additionally, we attempted to evaluate GSEDitPro as a baseline; however, we could not include it due to the unavailability of its official implementation. We intend to include GSEDitPro as a baseline in our evaluation once the official code is publicly released.
>
> ## [@LEXJ] - Explanation on Aligning Differently Shaped Feature Maps
>
> To address the varying feature shapes between DINO and diffusion features, we interpolate the ViT-based DINO features to match the dimensions of diffusion features. We will explicitly detail this interpolation procedure in the revised paper.
>
> ## [@LEXJ] - Clarification on Injection Layer Selection
>
> Our selection of specific layers for injecting diffusion features into Instruct-Pix2Pix was guided by insights from the original Plug-and-Play (PnP) [1] paper. Specifically, the PnP study conducted PCA visualizations across various layers within the Stable Diffusion v1 UNet architecture, demonstrating that the fourth layer of the UNet up-block effectively captures spatial information. Following this methodology, we verified through similar PCA visualizations that the fourth layer of the fine-tuned Instruct-Pix2Pix model similarly contains meaningful spatial information. Consequently, we chose this particular diffusion feature layer for feature injection, ensuring effective preservation and transfer of spatial structure during editing. We will clarify this selection process explicitly in the paper.
>
> [1] Tumanyan, Narek, et al. "Plug-and-play diffusion features for text-driven image-to-image translation." Proceedings of the IEEE/CVF conference on computer vision and pattern recognition. 2023.
>
> ## [@LEXJ] - Limitation of Dual Encoding
>
> Indeed, our method requires training an additional dual encoder and feature field specifically for each source scene prior to editing. This introduces extra computational overhead, with approximately 4 minutes required for feature field training. However, this step only needs to be conducted once per source scene and does not need repetition for subsequent editing tasks.
>
> Additionally, the complexity and structural details of scenes vary significantly across different samples, necessitating careful tuning of hyperparameters for optimal performance. Therefore, hyperparameter adjustments might be required on a per-sample basis to ensure the structure encoder accurately captures the appropriate level of structural detail. We acknowledge these limitations clearly and will explicitly include them in our paper.

---

> ### Author Response · Authors · 2025-08-05
>
> We sincerely thank the reviewer for carefully reviewing our rebuttal and acknowledge that your concerns have been largely addressed. We fully appreciate your point regarding the importance of supplementary video demonstrations. Unfortunately, due to the limitations of the rebuttal stage, we were unable to include supplementary videos. However, we commit to providing comprehensive video materials in the final manuscript, clearly demonstrating the effectiveness of our method.
>
> Additionally, we will carefully address your valuable comments on Incomplete Coverage of Related Work, Redundancy and Drawbacks of Aligning Diffusion Features with DINO, Justification for Multi-Timestep Usage, and Clarification on Injection Layer Selectionin the revised version.
> We deeply appreciate your valuable feedback and understanding.

---

### Official Review · Reviewer_MgqZ · 2025-07-02

**Clarity:** 2
**Significance:** 3
**Originality:** 3
**Rating:** 5
**Confidence:** 4

**Summary:**

The paper introduces DFFSplat, a pipeline for text-driven 3D scene editing in which both the source scene and the edited result are represented with 3D Gaussian Splatting (3DGS). The core idea is to learn a 3D-consistent diffusion feature field from multi-view diffusion features of the source scene and to inject these structural features back into a 2D diffusion editor (InstructPix2Pix) during per-view editing. A dual-encoder disentangles view-invariant structural features from view-dependent appearance, and a time-invariance regularizer makes the structural branch usable as a static 3D field. Editing is further steered by an additional view-consistency guidance term.

**Questions:**

## Questions & Suggestions

### Time Budget
I am confused by the claimed runtime of 4 minutes in Table 1, as a different number is mentioned in the supplemental material. Could you please clearly list all steps involved in your method—such as 3DGS training, feature field creation, dual-encoder inference, and editing—and indicate how much time each one takes in total?

### Limitations
You marked "Yes" in the checklist regarding a discussion of limitations, but I could not find any such section in the paper. Please explicitly list your method’s limitations, either in a separate section or clearly within the main text.

### Qualitative Consistency
The results in the main paper and supplemental differ significantly in visual quality. Could you clarify whether the same models, training setup, and hyperparameters were used for both? If not, what changed? Also are the results all single shots or multiple tries?

**Ethical Concerns:**

["NO or VERY MINOR ethics concerns only"]

**Final Justification:**

I recognize the concerns raised by my fellow reviewers. Judging a paper that emphasizes qualitative performance without another round of qualitative results is challenging. However, given that the authors are not responsible due to the NeurIPS technical issues and the rebuttal addresses the critiques convincingly, I am inclined to give them the benefit of the doubt.

I particularly like the base injection idea it seems both novel and potentially valuable for future work. The results, especially those in the supplementary materials, appear strong, and the runtime improvements are noteworthy. While the quantitative CLIP results remain somewhat inconclusive, they do not detract significantly from the paper’s overall contributions.

I would be willing to change my rating to accept, provided no other reviewer raises substantial concerns. That said, I strongly recommend the authors revise the Limitations, Related Work, and Runtime Analysis sections, and, if possible, include better qualitative results and a video demonstration.

**Limitations:**

I could not find any explicit discussion of the limitations of the proposed method in the main paper, even though the authors marked “Yes” in the checklist. Before attempting to list these myself, I would like the authors to clearly address this point and explicitly state the key limitations of their approach.

**Quality:**

3

**Strengths And Weaknesses:**

## Methodology

### Strengths
- Novel 3D diffusion feature field that explicitly enforces multi-view consistency.
- Clean dual-encoder design with separate structural and appearance branches.

### Weaknesses
- Feature field training and the auto-encoder introduce additional complexity. A detailed compute breakdown is missing.

---

## Evaluation

> Note: I initially considered giving a much lower evaluation score, as the results shown in the main paper did not appear very significant to me.

### Strengths
- Supplementary results look visually pleasing.
- In some cases, the Janus problem appears to be mitigated.

### Weaknesses
- The main paper's results are not promising and only marginally better than prior work. For example, the panda remains partly grey, Batman has double ears (Fig. 3), and the Spiderman mask is on par with previous methods. Overall benefits remain subtle.
- I would have expected some video material in the supplemental to illustrate view-dependent effects and multi-view consistency.

---

## Writing & Clarity

### Strengths
- The paper is generally well-structured.
- Mathematical formulations are explicit.

### Weaknesses
- The related work section is short and lacks depth.

---

## Impact

### Strengths
- Tackles a high-value problem: reliable text-driven 3D editing.
- The idea of merging diffusion features in 3D is promising and could inspire other use cases.

### Weaknesses
- The actual total compute time is unclear; see related question.

---

## Additional Comment

I appreciate the core insight and find the idea of merging diffusion features in 3D highly interesting. Before reviewing the supplemental material, I was not convinced that DFFSplat delivers meaningfully better results than state-of-the-art NeRF or GS editors. The improvements shown in the main paper are small and occasionally even negative.

However, the supplemental results are notably stronger, and I urge the authors to revise the main paper to match this quality. Currently, the main paper and supplemental feel disconnected. I also find the related work section too brief—it omits relevant methods that project features into 3D, especially from the scene understanding literature.

Finally, although the authors use GS as their representation, they should still compare against other recent methods such as IN2N and SIGNeRF.

I will give a **weak accept** for now because I like the core idea and want to acknowledge the promising results shown in the supplemental.

---

> ### Author Rebuttal · Authors · 2025-07-31
>
> ## [@MgqZ] - Clarity in Runtime and Pipeline Complexity
>
> Thank you for highlighting the confusion regarding runtime complexities in reported times. The average runtime indicated in Table 1 specifically refers to the editing phase itself and does not include the feature field creation step. We apologize for any confusion this may have caused.
>
> Our methodology involves several preliminary steps prior to the editing process. First, a dual-encoder is trained, which takes approximately 30 seconds. Subsequently, training the 3D diffusion feature field requires an additional 4 minutes. Importantly, this 3D-consistent diffusion feature field needs to be generated only once for a given source scene and does not have to be recreated for each editing task. After these steps are completed, the actual text-driven 3D editing operation takes approximately 4 minutes per task.
>
> To summarize, the complete breakdown of the computational steps involved is as follows:
>
> - Dual-encoder training: ~30 seconds (one-time)
> - 3D diffusion feature field training: ~4 minutes (one-time)
> - Text-based editing step: ~4 minutes (per editing instance)
>
> We will clearly specify these details in the paper to avoid any future confusion.
>
> ## [@MgqZ] - About Qualitative Improvements
>
> Regarding the observed limitations and subtle improvements compared to prior work, we appreciate your critical observations. To clearly illustrate the resolution of the Janus problem and semantic inconsistency issues with the source image, our main paper deliberately presented challenging cases. Most baseline methods failed to effectively address these challenging scenarios, whereas our proposed method successfully resolved them.
>
> Additionally, to demonstrate the capability of our approach to address these challenges within a limited number of training iterations, we intentionally restricted the training iterations for editing 3DGS to approximately 1000 iterations during our experiments. Consequently, some results in paper might appear less promising. However, allocating additional training time resolves issues such as "the panda remaining partly grey" and "Batman having double ears". Furthermore, increasing the text guidance scale beyond our experimental settings resulted in significantly improved performance.
>
> Additionally, we agree that including video materials illustrating multi-view consistency would substantially strengthen our evaluation. Therefore, we intend to incorporate comprehensive video examples in the supplementary material to clearly demonstrate these aspects and further validate the advantages of our approach.
>
> ## [@MgqZ] - Missing Discussion of Method Limitations
>
> We appreciate your comment regarding the explicit discussion of limitations. Initially, the limitations were provided in the supplementary material, but we agree that clearly listing them within the main text would improve readability and transparency. We will incorporate a dedicated limitations section into the main paper in our subsequent revision.
>
> ## [@MgqZ] - Inconsistency Between Main Paper and Supplemental Results
>
> We appreciate your careful observation regarding the visual quality differences between the main paper and supplementary results. For generating the supplementary results, we used the same dual-encoder and 3D feature field models as in the main paper. However, during the editing phase, we adjusted the number of iterations and the text guidance scale according to each scene's complexity. Typically, more complex scenes benefit significantly from a higher number of training iterations, resulting in enhanced editing performance. Additionally, increasing the text guidance scale leads to improved fidelity to the input text prompt.
>
> Interestingly, while increasing the guidance scale tends to exacerbate the Janus problem in baseline methods due to the property of stable diffusion model to generate predominantly front-view content, our method consistently demonstrates improved results without encountering the Janus issue.
>
> We agree with your suggestion to improve the visual quality presented in the main paper, as this will strengthen our paper. Thank you for highlighting this valuable point.
>
> ## [@MgqZ] - Lack of Related Works
>
> Thank you for highlighting the brevity and limited depth of our related work section. We agree that expanding and deepening this section will better position our contribution within the existing literature. We will enhance the related work section by thoroughly discussing additional relevant methods, clearly articulating our method's differences, and better contextualizing its novelty in the revised paper.
>
> ## [@MgqZ] - More Comparisons with Recent Baselines
>
> We have conducted additional comparisons and observed that our method significantly outperforms these NeRF-based baselines. While these experiments were performed under the same conditions as described in our main paper, we note that due to time constraints, our evaluations were limited to three scenes ("bear," "face," and "person"). We chose these specific scenes since they were originally employed in the DGE[1] paper and are thus well-known.
>
> | **Method** | **CLIP** | **CLIP Dir.** | **Time** |
> | --- | --- | --- | --- |
> | IN2N | 21.77 | 9.08 | ~15min |
> | SIGNeRF | 21.42 | 11.72 | ~15min |
> | Ours (Full method) | **25.22**  | **17.75**  | **~4min** |
>
> [1] Chen, Minghao, Iro Laina, and Andrea Vedaldi. "Dge: Direct gaussian 3d editing by consistent multi-view editing." *European Conference on Computer Vision*. Cham: Springer Nature Switzerland, 2024.

---

> ### Comment · Reviewer_MgqZ · 2025-08-03
>
> I thank the authors for their well-structured and clearly written rebuttal addressing all reviewers.
>
>
> ## Runtime
> The time constraints mentioned are interesting, yet the method still falls within a fast workflow approach, especially when compared to others that take significantly longer. Additionally, the ability to prepare one scene and generate multiple outputs within 4 minutes could be considered a valuable feature.
>
> ## Qualitative Improvements
> It is notable that extending training time resolves most issues, including the Janus problem. However, I remain skeptical as to why the authors did not explore this earlier, given that the longer training still fits within a reasonable timeframe. This point also appears vague compared with Reviewer (FBKe C3)’s comments on the Janus issue, why should this suddenly be solved with more iterations?
>
> ## NeRF Comparison
> I appreciate the inclusion of comparisons to existing NeRF methods. Still, I would have preferred to see qualitative comparisons as requested by Reviewer (WiaQ), since in my experience, quantitative CLIP scores can be vague and may not fully reflect the actual visual improvements. This could mask the benefits of the proposed Dual Encoder approach, as noted by Reviewer (FBKe).
>
> ## Overall Feedback
> I recognize the concerns raised by my fellow reviewers. Judging a paper that emphasizes qualitative performance without another round of qualitative results is challenging. However, given that the authors are not responsible due to the NeurIPS technical issues and the rebuttal addresses the critiques convincingly, I am inclined to give them the benefit of the doubt.
>
> I particularly like the base injection idea it seems both novel and potentially valuable for future work. The results, especially those in the supplementary materials, appear strong, and the runtime improvements are noteworthy. While the quantitative CLIP results remain somewhat inconclusive, they do not detract significantly from the paper’s overall contributions.
>
> I would be willing to change my rating to **accept**, provided no other reviewer raises substantial concerns. That said, I strongly recommend the authors revise the Limitations, Related Work, and Runtime Analysis sections, and, if possible, include better qualitative results and a video demonstration.

---

> > ### Author Response · Authors · 2025-08-04
> >
> > Thank you for recognizing the efficiency of our method. We appreciate your thoughtful consideration regarding the practical workflow and runtime advantages of our approach. We will thoroughly enhance the **Limitations**, **Related Work**, and **Runtime Analysis** sections to effectively address your remaining concerns. Additionally, we will provide comprehensive qualitative results, including comparisons with additional baselines, to better demonstrate the effectiveness of our method. Thank you once again for your valuable feedback.
> >
> > ## **Qualitative Improvements**
> >
> > To address your question regarding why additional iterations can resolve the Janus issue: We clarify that most existing methods, including ours, typically adopt an iterative pipeline of "view image editing → 3DGS optimization". In such a pipeline, multi-view inconsistencies arising during view image editing are gradually refined through subsequent 3DGS optimization steps, improving coherence across views. Thus, simply increasing the number of 3DGS optimization iterations alone does not directly resolve the issue; rather, the iterative refinement between these two processes gradually mitigates it.
> >
> > However, existing methods typically require significantly more iterations to alleviate multi-view inconsistency, resulting in substantially longer editing times. Moreover, even with increased iterations, the Janus problem may persist under challenging conditions, such as in scenarios with very sparse views or highly complex scenes. These conditions often induce frontal biases, hindering semantic consistency with the source. In contrast, our method directly injects cross-view consistent and semantically consistent information during the editing stage, explicitly addressing inconsistencies with fewer iterations. We will explicitly clarify this point in the revised manuscript.

---

### Official Review · Reviewer_FBKe · 2025-07-13

**Clarity:** 3
**Significance:** 3
**Originality:** 3
**Rating:** 4
**Confidence:** 4

**Summary:**

This paper proposes DFFSplat, a method designed to address multi-view inconsistency in the InstructGS2GS 3D editing pipeline, particularly the Janus problem. The core idea is to disentangle structural and appearance features from the intermediate features of a diffusion model using an autoencoder architecture. The structural feature is regularized by enforcing its self-similarity map to match that extracted by DINOv2. These structural features are then used to construct a 3D feature field, which enforces consistency across different views. At inference, the rendered structural feature is combined with the appearance feature and passed to the diffusion model for final image generation. Experiments show that this approach mitigates the Janus problem caused by view inconsistency.

**Questions:**

1. Please provide a more thorough ablation study to justify the role of the 3D feature field, including results averaged over five different random seeds with standard deviations.
2. Include an ablation study where both the 3D feature field and the appearance encoder $E_{\text{app}}$ are removed. This would help clarify the necessity and contribution of each component.
3. Please discuss the relevance of GaussCtrl in relation to your work, and provide a direct comparison or justification for its exclusion as a baseline.
4. Consider proofreading the manuscript to correct grammatical errors and clarify confusing sentences, particularly in Lines 52–56 and other areas mentioned in the review.

**Ethical Concerns:**

["NO or VERY MINOR ethics concerns only"]

**Final Justification:**

Thanks to the additional results and information provided by the authors, my concern about the necessity of the 3D feature field and appearance encoder has been largely resolved. Therefore, I would raise my score to "borderline accept".

**Limitations:**

Yes

**Paper Formatting Concerns:**

No formatting concern.

**Quality:**

3

**Strengths And Weaknesses:**

**Strengths**
1. The paper identifies the root cause of the Janus problem as the frontal-view bias in diffusion models and proposes to address this by disentangling the structural feature map and regularizing it using the self-similarity map from DINOv2. This design encourages the structural representation of rear views to differ from the frontal-view representation, thus helping to avoid Janus artifacts. The motivation and intuition behind this approach are well articulated and logically sound.
2. The paper is generally well-written and easy to follow, aside from a few grammatical errors and unclear sentences, which I outline in the Weaknesses section.

**Weaknesses**\
I have two main concerns: one regarding the necessity and justification of the 3D feature field, and the other about the omission of an important baseline, GaussCtrl [1].

Necessity of the 3D Feature Field:
1. As discussed in the Strengths section, I believe the regularization of the structural feature via DINOv2 is the key factor in addressing the Janus problem. Once the structural features are regularized, the additional step of constructing a 3D feature field seems redundant. In fact, as acknowledged by the authors in Line 150, the feature field performs an averaging operation, which may dilute the fine-grained structural information. This weakens the justification for its inclusion in the pipeline.
2. The paper attempts to justify the 3D feature field via an ablation study in Table 2. However, the differences in CLIP and CLIP Dir metrics are minimal and within a range that could easily be attributed to randomness. Given that the InstructGS2GS pipeline is known to be highly stochastic, these results do not provide convincing evidence. A more robust experiment would be to repeat the evaluations across five different random seeds and report the mean and standard deviation of the metrics.
3. The plot on the right side of Figure 4 shows increased feature distortion when using the 3D feature field, which further reinforces my concern that this module may be doing more harm than good.
4. I also question the role of the appearance encoder $E_{\text{app}}$. Typically, disentangling tasks, especially in the context of appearance vs. structure, require explicit supervision. An example is [2], where pose and shape attributes are disentangled with appropriate supervision. In contrast, $E_{\text{app}}$ is trained without supervision, raising doubts about whether it genuinely captures appearance information. Furthermore, since $E_{\text{str}}$ is supervised only via the DINOv2 self-similarity map (which may already encode appearance to some extent), it is unclear whether $E_{\text{app}}$ is necessary at all. It may be possible for a single encoder to learn a sufficiently disentangled representation on its own.

Missing Baseline:
1. In Line 35 of the introduction, the paper suggests that existing methods struggle with multi-view consistency, potentially implying that this work is the first to identify or address the issue. However, multi-view inconsistency is a well-known and long-standing challenge in 3D editing. A particularly relevant and recent work is GaussCtrl [1], which addresses geometric and appearance consistency by conditioning editing on both a reference view and a depth map. Its motivation is closely aligned with that of this paper, and it specifically claims to reduce the Janus effect. Despite its relevance, GaussCtrl is neither cited nor used as a baseline in this work, which is a significant omission.

Minor Errors:
1. Line 52: "related" → "relates"
2. Lines 54–56: The sentence is awkward and grammatically incorrect. "Isolating" should be paired with "from," not "into." The sentence should be rephrased for clarity.
3. Equation 4 indicates that the encoder takes a timestep $t$ as input, but this dependency is not reflected in Figure 2.

**Conclusion**\
Due to the concerns outlined above, particularly the unclear necessity of the 3D feature field and the omission of a key related method (GaussCtrl), I currently recommend Reject. However, I am open to revisiting this recommendation if the authors can adequately address the points raised in this review.

[1] Wu, Jing, Jia-Wang Bian, Xinghui Li, Guangrun Wang, Ian Reid, Philip Torr, and Victor Adrian Prisacariu. "Gaussctrl: Multi-view consistent text-driven 3d gaussian splatting editing." In European Conference on Computer Vision, pp. 55-71. Cham: Springer Nature Switzerland, 2024.\
[2] Zhou, Keyang, Bharat Lal Bhatnagar, and Gerard Pons-Moll. "Unsupervised shape and pose disentanglement for 3d meshes." In European conference on computer vision, pp. 341-357. Cham: Springer International Publishing, 2020.

---

> ### Author Rebuttal · Authors · 2025-07-31
>
> ## [@FBKe] - On the Necessity of the 3D Feature Field
>
> We appreciate Reviewer FBKe's thoughtful comments and would like to clarify the motivation and necessity of the proposed 3D feature field beyond the regularization of structural features using DINOv2.
>
> While the DINO-based regularization indeed helps extract semantically meaningful and spatially coherent structural features, we argue that the 3D feature field plays an indispensable role in enforcing structural consistency across views, which cannot be achieved by regularization alone. Specifically, we highlight three key justifications:
>
> ### Viewpoint-Agnostic Semantic Fusion:
> As stated in Sec. 4.1–4.2 of our paper, the 3D feature field aggregates diffusion features across multiple views into a unified representation. This operation not only removes view-specific noise but also harmonizes structural discrepancies caused by the diffusion model’s frontal-view bias (see Line 38–43). While this averaging process may appear lossy, it effectively neutralizes conflicting semantic cues across views and ensures geometric alignment at the feature level.
>
> To evaluate this point, we conducted an ablation study assessing multi-view and semantic consistency via correspondence matching. Specifically, as mentioned in the paper, we compute pixel-level correspondence matching maps among multi-view images rendered from both the original and edited 3D scenes, and then measure the differences between these maps. A smaller difference indicates better preservation of the source scene’s geometry and improved multi-view consistency due to accurate pixel-level matching across views. As demonstrated in the table below, incorporating the 3D feature field consistently yields improved multi-view and semantic consistency. Please note that these results are computed as the mean across five random seeds.
>
> | Method | Difference in correspondence matching |
> | --- | --- |
> | Ours | **3.19** |
> | - 3D feature field | 3.66 |
>
> ### Interpretation of Feature Distortion in Fig. 4:
> Reviewer FBKe points out the increased distortion in Fig. 4 when using the 3D feature field. However, we argue that this result does not reflect degradation, but rather a necessary correction. The increased distortion stems from our 3D field overriding incorrect features (e.g., Janus artifacts caused by frontal bias) to enforce consistent multi-view structure. As further evidence, this distortion trend remains stable across time steps (if substantial fine-grained information had been lost, we would expect a larger distortion at low-noise timesteps), where appearance cues dominate (see Fig. 4 Right). The consistency of the distortion across timesteps supports our claim that the 3D field preserves fine-grained appearance via $E_{app}$ while prioritizing semantic alignment.
>
> ### Empirical Improvements & Visual Coherence:
>
> Although the numerical gain in CLIP metrics shown in Table 2 may appear modest, the visual improvements are pronounced, especially in challenging cases (e.g., the panda face on the back in Fig. 1 & Fig. 3). These qualitative results highlight that the 3D feature field mitigates Janus-like inconsistencies that neither the structure encoder nor appearance encoder alone can address. We also acknowledge the stochasticity of the editing process and are preparing multi-seed results as the reviewer suggested, which will be included in the revised supplementary.
>
> ---
> In summary, the 3D feature field is not simply a redundant addition to the DINO-regularized encoder but rather a necessary mechanism to fuse multi-view semantic structure and eliminate artifacts rooted in the inherently 2D nature of diffusion editing. Without this explicit 3D fusion, rear-view inconsistencies and view conflicts persist (as confirmed in our ablations in Fig. 4 and Table 2).
>
> ## [@FBKe] - Role and Necessity of the Appearance Encoder
>
> We appreciate the reviewer’s careful concern of $E_{app}$. Below we clarify its motivation, training signal, and empirical contribution.
>
> ### Separating What to Keep vs. What to Restore
>
> Our structural branch ($E_{str}$) is *aggressively* regularised: (1) DINOv2 self-similarity aligns global geometry, and (2) the time-invariance loss collapses features across *all* timesteps into a single 3D field. This design intentionally *filters out* view-dependent, high-frequency cues that will be edited (i.e., removed). $E_{app}$ is therefore indispensable for *re-injecting* those cues within the editing process; otherwise the edited outputs become over-smoothed (Fig. 4. left) and appearance information from prompt-conditions cannot be properly applied.
>
> Here we would like to re-emphasize that this is enabled by our carefully designed method that jointly learns $E_{app}$ (via a dual-encoder scheme) and the 3D feature field, allowing precise and controllable manipulation over 3D visual attributes. Importantly, such disentanglement is not optional but rather mandatory for effective 3D editing. For these reasons, we respectfully disagree that $E_{app}$ is redundant. It is a lightweight yet crucial component that (1) safeguards fine details, (2) enlarges the editing expressiveness of DFFSplat through disentangled control.
>
> We sincerely appreciate the time and effort for the insightful review.
>
>
> ### Synergy with the 3D Feature Field
>
> Above, we have discussed the significant benefits and necessity raised by 3D field.
> However, this only holds with the assumption that our $E_{app}$ is accompanied.
> Without this, Fig.4 shows that it indeed leads to non-neglectable information loss.
> Thus, in order to fully retain the advantages of 3D field while not suffer from significant info loss, it must be jointly utilized.
>
> Below, we provide more specified discussion.
>
> Fig. 4 (right) shows that the apparent distortion introduced by 3D fusion *plateaus* when $E_{app}$ is present, whereas it *grows* with timestep if $E_{app}$ is removed.
> This indicates that the view-specific noise/texture are reconstructed via the $E_{app}$-relevant path, which enables the 3D field to purposefully discard these information (beneficial in editing).
>
>
> ### Single-Encoder Variant Fails to Disentangle
>
> The dual-encoder shares a common decoder and is trained with an overall reconstruction loss. Because $E_{str}$ is tightly constrained, any residual information needed to minimize that loss *must* flow through $E_{app}$. This “information-bottleneck” effect acts as an implicit appearance supervision.
>
> As discussed above the informational bottlneck designed via the _dual_-encoder is necessary.
> We implemented the reviewer’s suggestion (one encoder + DINO loss). Across **5 seeds** the average score show significant drop. Also visually, Janus artifacts resurfaced (panda eyes on bear back; Batman double-ear).
>
> Unfortunately, visual examples are not allowed to be presented here.
> The ablation analysis and visual results will be added in our final manuscript.
>
> | Method | CLIP (mean / std) | clip_dir (mean / std) |
> | --- | --- | --- |
> | Ours (Full Method) | 25.13 / 0.034 | 17.94 / 0.078 |
> | - 3D feature field | 25.09 / 0.037 | 17.87 / 0.226 |
> | - $E_{app}$ | 24.42 / 0.254 | 16.79 / 0.491 |
>
> [1] Yue, Zhongqi, et al. "Exploring diffusion time-steps for unsupervised representation learning." *arXiv preprint arXiv:2401.11430* (2024).
>
> [2] Tumanyan, Narek, et al. "Splicing vit features for semantic appearance transfer." Proceedings of the IEEE/CVF Conference on Computer Vision and Pattern Recognition. 2022.
>
> ## [@FBKe] - Missing Baseline
>
> Thank you for highlighting the omission of the GaussCtrl baseline. GaussCtrl addresses geometric and appearance consistency on multi-view by conditioning edits on a reference view and a corresponding depth map. However, we observed that relying solely on depth maps often fails to capture sufficient semantic information, leading to semantic inconsistencies with the source scene. For instance, in our "bear-to-panda" experimental scenario, GaussCtrl generated inappropriate semantic artifacts, such as facial features incorrectly appearing on the panda's backside, thus exacerbating the Janus problem. This indicates that depth maps alone are insufficient for robust semantic guidance. In contrast, our method effectively maintains semantic consistency by the guidance of structure features from source scene.
>
> Additionally, quantitative comparisons demonstrated that our approach achieves higher editing fidelity and faster inference speeds compared to GaussCtrl. We will include these detailed comparative experiments and an expanded discussion of GaussCtrl in our revised paper to clarify its relation to our proposed method.
>
> | Method | CLIP | CLIP Dir. | Time |
> | --- | --- | --- | --- |
> | GaussCtrl | 23.77 | 15.25 | ~8min |
> | Ours  | **25.29** | **17.69** | ~4min |
>
> [1] Wu, Jing, et al. "Gaussctrl: Multi-view consistent text-driven 3d gaussian splatting editing." European Conference on Computer Vision. Cham: Springer Nature Switzerland, 2024.

---

> ### Comment · Reviewer_FBKe · 2025-08-03
>
> I thank the authors for their detailed response. I would like to make the following comments:
>
> - I am now convinced that the appearance encoder is an indispensable part of the pipeline, especially through the ablation study that removes the decoder. The performance drop is substantial and supports the value of the encoder. I sincerely thank the authors for their effort in providing this result.
>
> - However, my reservation on the 3D feature field remains, and my reasons are:
>
> 1. From the second table in the rebuttal, the difference between keeping and removing the 3D feature field is very small, especially considering the std. Such a small performance gain may not fully justify the use of the 3D feature field.
> 2. Regarding the correspondence experiment, few details have been given. How is this correspondence map generated? Is it a dense correspondence map or the map of sparse keypoints? How to make sure that the difference is from the multiview inconsistency, but not from the undesired change from the background?
> 3. Regarding the feature distortion, I doubt that "increased distortion stems from our 3D field overriding incorrect features". The comparison is between the reconstructed feature $\hat{F}^t$ from the decoder at the training stage (equation 4) and the modified feature $\hat{F}^t_{injected}$ at the inference stage (equation 9). In the ideal case, they are supposed to be the same. The frontal bias is supposed to already be corrected by the optimization with DINO. Therefore, I do not believe any correction of the Janus problem is involved here. Moreover, I am not indicating that using the 3D feature field will cause the substantial loss of fine-grained details, but merely stating the fact that Figure 4 shows the 3D feature field will make the feature distortion slightly worse.
> 4. Conceptually, the 3D feature field has no ability to correct the Janus problem. The alignment to the DINO feature does this work. I acknowledge that it may smooth features across different views and suppress the noise to some extent. However, its inclusion requires more justification, and does the benefit truly outweigh the cost?
>
> - I thank the authors for considering GaussCtrl and providing relevant results.

---

> ### Author Response · Authors · 2025-08-04
>
> ## Details of the correspondence experiment
>
> First, we edit rendered images and then extract features from these edited images using DINOv2 (ViT-L/14), specifically utilizing layer 23 token features. Subsequently, we employ Language-SAM to obtain object-specific masks, which we use to mask out background regions from these features. Utilizing these masked features, we calculate dense correspondence maps by identifying nearest patches across all pairs of view images. We then compute the differences between the correspondence maps of the original and edited images, measuring these differences using the L2 distance.
>
> By explicitly masking out background regions during this calculation, we ensure that our reported differences reflect changes in object regions rather than unintended background alterations.
> This score measures multi-view consistency in edited images by comparing their correspondence maps with those of inherently consistent source images, simultaneously indicating how effectively the original 3D geometric structure is preserved.
>
> ## 3D feature field for mitigating Janus problem
>
> We agree that regularization using DINO self-similarity maps can partially mitigate the Janus problem. However, we emphasize that the 3D feature field is still crucial, as relying solely on DINO regularization does not fundamentally correct the frontal bias inherent in diffusion features. Specifically, the structure encoder trained via DINO self-similarity maps primarily works to disentangle structure and appearance, rather than explicitly correct the frontal bias. It is challenging for this approach to selectively exclude features exhibiting frontal bias while including only unbiased features.
>
> We attribute this to two primary reasons: (1) our dual-encoder network is too shallow, designed only to encode structural information; and (2) since frontal biases occur infrequently, the encoder naturally learns generalized structural information. For these reasons, with DINO regularization, the structure encoder lacks the sophisticated capability to correct frontal bias. Consequently, without passing through the 3D feature field, the structure features alone still exhibit frontal bias. Evidence supporting this is clearly demonstrated in below table, which shows that the presence or absence of DINO regularization does not significantly impact correspondence matching scores.
>
> In contrast, the 3D feature field can effectively mitigate frontal biases. Notably, frontal biases are inherently (1) inconsistent across different 3D views and (2) relatively rare within the overall set of viewpoints. Therefore, by explicitly averaging structural information across multiple views, the 3D feature field can correct biased views by utilizing structural information from unbiased views. This capability of cross-view averaging offered by the 3D feature field effectively resolves these biases, an advantage not attainable with DINO regularization alone.
>
> To further analyze the individual impacts of DINO regularization and the 3D feature field, we conducted the correspondence matching experiment by selectively removing each component. Specifically, we compared scenarios without DINO regularization and without the 3D feature field, respectively. Since the appearance encoder ($E_{app}$) cannot be trained effectively without DINO regularization, we performed the comparative experiments excluding the appearance encoder.
>
> |  | Difference in correspondence matching |
> | --- | --- |
> | w/o $E_{app}$ | 2.7022 |
> | w/o $E_{app}$ + w/o DINO regularization | 2.7253 |
> | w/o $E_{app}$ + w/o 3D feature field | 4.2175 |
>
> From the table above, we observe that excluding the 3D feature field results in a significantly larger difference in the correspondence matching map compared to excluding DINO regularization. This clearly indicates that the 3D feature field substantially contributes to improving multi-view consistency and semantic consistency, rather than DINO regularization.
>
> ## About small performance gain in CLIP of the use of the 3D feature field
>
> As demonstrated in the second table of our rebuttal, the 3D feature field does not significantly affect edit fidelity as measured by CLIP scores. However, CLIP scores, which only evaluate alignment between images and edit prompts, cannot adequately capture the multi-view and semantic consistency we aim to achieve. Thus, we emphasize the correspondence matching scores, which better reflect the extent of the Janus problem. Indeed, these scores (as presented in the first table of our rebuttal) clearly show that the 3D feature field substantially enhances multi-view and semantic consistency, effectively mitigating the Janus problem, despite its limited impact on CLIP scores.
>
> ---
>
> We hope our responses have clearly addressed your concerns. Your insightful comments allowed us to conduct more detailed analyses and further clarify the contributions of our method.

---

> > ### Comment · Reviewer_FBKe · 2025-08-05
> >
> > I thank the authors for their detailed response.
> >
> > So it seems that the correspondence map difference can measure the Janus problem better. Based on my understanding, this metric treats the DINO correspondence of the original multiview images as the ground truth and compares DINO correspondence of the edited multiview images against it. It makes sense as long as DINO can correctly reflect the geometry of the object. In my opinion, this is an important metric and should serve alongside or even take over the CLIP as the main metric of the experiment, instead of being used as a supplementary because one of the key claims of the paper is to reduce the Janus problem.
> >
> > I am now convinced about the use of the 3D feature field, and I will update my score to borderline accept. I recommend the authors include the following changes to the final version:
> > 1.  Treat correspondence map difference as one of the main metrics of the experiment, clearly state its definition and its motivation which is to measure the Janus problem.
> > 2. Include all additional results in our discussion to the experiment.
> > 3. More explanation on the necessity of 3D feature field in alleviating the Janus problem. I believe the intuition
> >
> > *the 3D feature field can effectively mitigate frontal biases. Notably, frontal biases are inherently (1) inconsistent across different 3D views and (2) relatively rare within the overall set of viewpoints. Therefore, by explicitly averaging structural information across multiple views, the 3D feature field can correct biased views by utilizing structural information from unbiased views*
> >
> > is useful and should be clearly marked in the main paper. Moreover, more visualization on the difference between adding and removing 3D feature field may also help.
> >
> > In the end, I thank the authors again for all their efforts in providing additional results and insightful discussions on my concerns.

---

> > > ### Author Response · Authors · 2025-08-05
> > >
> > > We sincerely thank the reviewer for their thoughtful consideration and detailed comments.
> > >
> > > We greatly appreciate your acknowledgment that the correspondence map difference metric effectively captures the Janus problem. Given its ability to accurately measure multi-view and semantic consistency, central objectives of our paper, we fully agree that this metric should be highlighted more prominently.
> > >
> > > In line with your recommendations, we will implement the following revisions in our final manuscript:
> > >
> > > - Clearly introduce the correspondence map difference as a primary evaluation metric, explicitly defining its purpose and emphasizing its suitability for robustly assessing the Janus problem.
> > > - Incorporate all additional results and analyses presented during this rebuttal phase to further strengthen and clarify our methods.
> > > - Provide detailed explanations on the necessity of the 3D feature field in addressing the Janus problem. Specifically, we will highlight that frontal biases are inherently (1) inconsistent across different 3D viewpoints and (2) infrequent among views, enabling the 3D feature field to effectively mitigate these biases by averaging structural information from unbiased viewpoints.
> > > - Include visualizations explicitly illustrating differences in results obtained with and without the 3D feature field, thereby clearly demonstrating its crucial role in resolving the Janus issue.
> > >
> > > We deeply appreciate your valuable insights and constructive suggestions, which have significantly enhanced the clarity and robustness of our manuscript.

---

### Note · Authors · 2025-08-14

We thank all reviewers and the AC for the constructive discussions, which offered valuable insights and enabled us to further strengthen our work. Below, we provide the key points discussed during the rebuttal phase.

**1. Necessity of the 3D Feature Field**

In response to Reviewer FBKe, we clarified why the 3D feature field is essential for addressing the Janus problem. This issue arises when the diffusion model’s frontal bias appears in non-frontal views during per-view editing. Because frontal bias (1) occurs inconsistently across views and (2) is relatively rare, cross-view averaging in the 3D feature field can refine biased views using unbiased structural information from other viewpoints. Our correspondence matching experiment demonstrated that the 3D feature field effectively enhances multi-view and semantic consistency, which is crucial for addressing the Janus problem. The results showed that removing the 3D feature field increased the correspondence difference from 2.70 to 4.21, far greater than the change from removing DINO regularization (2.70 to 2.72), thereby demonstrating its critical contribution and indispensable role.

**2. Role of the Appearance Encoder**

Following Reviewer FBKe’s request, we conducted an ablation study without the appearance encoder, which led to substantial performance degradation. This confirms that the appearance encoder is indispensable for maintaining editing fidelity.

**3. More Baselines and Related Work**

As suggested by Reviewers FBKe, MgqZ, and LEXJ, we will expand our discussion to include GaussCtrl, IN2N, SIGNeRF, and GSEditPro (once code becomes available). Comparative results against GaussCtrl and other NeRF-based baselines presented in the rebuttal demonstrate the superior fidelity and efficiency of our method.

**4. Structural Editing Capability**

In response to Reviewer WiaQ, we demonstrated that our framework can be extended to object removal and addition tasks by leveraging the semantic richness of the 3D feature field, highlighting the broader applicability and potential of our approach.

**5. Additional Materials**

Multiple reviewers emphasized the importance of supplementary videos and a broader range of results. We will incorporate these into the final manuscript to provide stronger qualitative evidence of our method’s effectiveness.

We once again thank all reviewers for their valuable comments, which have been instrumental in improving the clarity and strength of our manuscript.

---

### Decision · Program_Chairs · 2025-09-17

**Decision:**

Accept (poster)

**Comment:**

Majority of the reviewers are leaning towards accept for this paper. Reviewers appreciated the well-written paper with interesting technique to disentangle the appearance and structure to mitigate multi-view inconsistency issues. On the other hand, multiple reviewers raised concerns related to missing comparisons, experiments and visual results. AC carefully looked at the reviews and the author response. Authors addressed several of the concerns in the rebuttal. It is felt that the strengths outweigh the weaknesses. The reviewers did raise some valuable concerns that should be addressed in the final camera-ready version of the paper, which include adding the relevant rebuttal discussions in the main paper. The authors are encouraged to make the necessary changes to the best of their ability, especially it is important to present video results with the work.